# Random Walks on Comb-like Structures under Stochastic Resetting

**DOI:** 10.3390/e25111529

**Published:** 2023-11-09

**Authors:** Axel Masó-Puigdellosas, Trifce Sandev, Vicenç Méndez

**Affiliations:** 1Grup de Física Estadística, Departament de Física, Universitat Autònoma de Barcelona, Edifici Cc, E-08193 Cerdanyola, Spain; amaso@crm.cat; 2Research Center for Computer Science and Information Technologies, Macedonian Academy of Sciences and Arts, Bul. Krste Misirkov 2, 1000 Skopje, Macedonia; trifce.sandev@manu.edu.mk; 3Institute of Physics & Astronomy, University of Potsdam, D-14476 Potsdam, Germany; 4Institute of Physics, Faculty of Natural Sciences and Mathematics, Ss Cyril and Methodius University, Arhimedova 3, 1000 Skopje, Macedonia

**Keywords:** anomalous diffusion, random walks, stochastic resetting

## Abstract

We study the long-time dynamics of the mean squared displacement of a random walker moving on a comb structure under the effect of stochastic resetting. We consider that the walker’s motion along the backbone is diffusive and it performs short jumps separated by random resting periods along fingers. We take into account two different types of resetting acting separately: global resetting from any point in the comb to the initial position and resetting from a finger to the corresponding backbone. We analyze the interplay between the waiting process and Markovian and non-Markovian resetting processes on the overall mean squared displacement. The Markovian resetting from the fingers is found to induce normal diffusion, thereby minimizing the trapping effect of fingers. In contrast, for non-Markovian local resetting, an interesting crossover with three different regimes emerges, with two of them subdiffusive and one of them diffusive. Thus, an interesting interplay between the exponents characterizing the waiting time distributions of the subdiffusive random walk and resetting takes place. As for global resetting, its effect is even more drastic as it precludes normal diffusion. Specifically, such a resetting can induce a constant asymptotic mean squared displacement in the Markovian case or two distinct regimes of subdiffusive motion in the non-Markovian case.

## 1. Introduction

Stochastic resetting has been a recent field of study in the physical literature. Since the first work devoted to study diffusion under Markovian resetting [1], it has been deeply analyzed when applied to different stochastic processes [2,3,4,5]. Recently, interest in diffusion in heterogeneous media under resetting has increased [6,7,8,9,10,11,12,13,14,15,16,17]. A common feature of these works is that the motion of the walker through the media is subdiffusive, which means that the mean squared displacement (MSD) scales as $\langle x^{2}\left(t\right)\rangle ∼t^{\alpha }$, 0<α<1 [18]. A simple prototypical model of heterogeneous media is the comb-like geometry [19]. Combs are two-dimensional branched structures, with a backbone crossed by perpendicular fingers, which have been used in different contexts [20,21,22,23,24]. A random walker moving diffusively along the backbone may enter into a finger and move there for a time and return to the backbone to start moving there again. As a result, the MSD along the backbone shows a subdiffusive behavior depending on time scaling as $\langle x^{2}\left(t\right)\rangle ∼t^{1/2}$. When resetting is included, one may consider two different mechanisms. On one hand, the walker may reset its position to the backbone during the motion along the fingers [21]. On the other hand, the resetting mechanism may be global and hence the walker may reset its position to the origin regardless of its current position. These situations have been recently studied and both the MSD and the propagator of the overall process have been computed [6,9,25]. In these works, the motion along fingers and backbone is assumed to be diffusive. Here, we consider global and local resetting, in fingers, of a walker moving diffusively along the backbone but subdiffusively when it moves along the fingers. Subdiffusion along fingers is introduced by taking the probability density function (PDF) of the waiting time between short-distance jumps to be heavy-tailed [18]. This subdiffusion along the fingers may be a result of additional fingers on the fingers of the comb [6,26,27]. A new result here is also the introduction not only of a Poissonian resetting, but also a non-Markovian (power-law) resetting in the comb structure, which has not been considered elsewhere. To this end, we consider two coupled Langevin equations to describe the motion of the walker under global resetting or resetting in fingers. The resetting dynamics we consider here are taken to be heavy-tailed as well. We explore the interplay between the exponents of the waiting times and resetting event PDFs analytically.

## 2. A Langevin Equation Approach to Random Walks on Combs

The dynamics of a random walker moving on a comb can be described by the Langevin equations [28,29]
(1)$$\begin{array}{ccc} \frac{dX}{dt} & = & C\left(Y\right)\xi_{x}\left(t\right),\\ \frac{dY}{dt} & = & \xi_{y}\left(t\right), \end{array}$$
where {X(t),Y(t)} is a random process describing the position of a walker moving along the backbone (*x*-direction) according to X(t) and along the fingers (*y*-direction) according to Y(t). A comb model is a toy model of a non-Markovian motion which occurs due to the specific two-dimensional structure. It consists of a backbone along the *x* direction and continuously distributed fingers (or branches) along the *y* direction, as shown in Figure 1. The particle moving along the backbone is trapped in the fingers. The time spent in the fingers can be considered as a waiting time of the particle’s movement along the backbone. Therefore, the resulting motion along the backbone is anomalous. The motion along each direction is driven by the uncorrelated external noises $\xi_{x}\left(t\right)$ and $\xi_{y}\left(t\right)$. The noise $\xi_{x}\left(t\right)$ is assumed to be white and Gaussian with an autocorrelation function $\langle \xi_{x}\left(t\right)\xi_{x}\left(t^{\prime }\right)\rangle =2D_{x}\delta \left(t-t^{\prime }\right)$. The coupling between the motions along the *x* and *y* directions is described by the coefficient C(Y). As can be seen from Equation (Equation 1), the dynamics of the walker along the *y* direction are independent of the *x* coordinate which indicates that the PDF of the random process Y(t), namely, $P_{Y}\left(y,t\right)$, does not depend explicitly on *x*. The noise $\xi_{y}\left(t\right)$ is arbitrary and such that $P_{Y}\left(y,t\right)={\langle \delta \left(Y\left(t\right)-y\right)\rangle }_{\xi_{y}}$. Integrating Equation (Equation 1) and making use of the Stratonovich interpretation of the stochastic calculus, the MSD along the overall structure reads [29]
(2)$$\langle X{\left(t\right)}^{2}\rangle =2D_{x}\int_{0}^{t}dt^{\prime }\int_{-\infty }^{\infty }C{\left(y\right)}^{2}P_{Y}\left(y,t^{\prime }\right)dy.$$
The double average in the above equation means that the mean of $X{\left(t\right)}^{2}$ is computed over the realizations of both $\xi_{x}$ and $\xi_{y}$ noises. For a ramified comb structure, the coupling between the motions along *x* and *y* axis reduces to the point where the teeth cross the backbone (i.e., y=0) so that the coupling coefficient is such that $C{\left(y\right)}^{2}=\delta \left(y\right)$. Hence, from (Equation 2) the overall MSD is
(3)$$\langle X^{2}\left(t\right)\rangle =2D_{x}\int_{0}^{t}P_{Y}\left(y=0,t^{\prime }\right)dt^{\prime }.$$
This expression establishes an interesting dependence of the overall MSD on the propagator of the motion along fingers evaluated at y=0.

Here, we note that the comb models can be used to describe the anomalous diffusion in spiny dendrites [21,30,31], including the anomalous diffusion in Purkinje cells [32,33]. There are different generalizations of the comb model in which one can consider, for example, finite-length fingers, multiple backbones, fractal structure of the fingers and/or the backbones, see, for example [19,34], combs with finger lengths drawn from a power-law distribution [27,35], and random comb models [36], as well as different circular and branched spiral and circular structures [37,38,39]. The corresponding diffusive behavior along the backbone would depend on the distribution of fingers and/or backbones in the comb-like structure. Those fractal generalizations of the standard comb are not just abstract mathematical models, but can be useful for the description of anomalous transport through porous solid pellets with various porous geometries [40]. Another possible application of such fractal combs can be in the modelling of river basins with their often very ramified geometry [41,42]. The long-time retention data of tracers in water catchments reveal scaling exponents consistent with comb dynamics [43,44].

## 3. Random Walks on Combs under Resetting

### 3.1. Resetting Process

We consider two resetting protocols: (i) a return to the backbone if the walker is moving along a finger, namely, *resetting in fingers* and (ii) a return to the origin regardless of its position, namely, *global resetting*; see Figure 1. In both mechanisms, resetting events take place at a random time *t* drawn from the PDF $\varphi_{R}\left(t\right)$. Even though the resetting to the initial position is sudden and somehow physically unrealistic, there are currently experimental realizations of diffusion processes under stochastic resetting, showing a good agreement with the analytical results. In particular, experimental realizations of the first passage under resetting have been demonstrated by using holographic optical tweezers [45] or laser traps [46]. Moreover, the stochastic resetting can be a good idealization of a model of a diffusing particle in confining potentials, which can be stochastically switched on and off [47]. Moreover, the resetting in the fingers can mimic the confining potential applied along the fingers, which brings back the particle in the backbone of the comb [25], or the finite size effects of fingers [19].

The probability that the *n*-th resetting event happens at time *t*, $\varphi_{R}^{\left(n\right)}\left(t\right)$ satisfies the renewal equation [48,49]
$$\varphi_{R}^{\left(n\right)}\left(t\right)=\int_{0}^{t}\varphi_{R}^{\left(n-1\right)}\left(t^{\prime }\right)\varphi_{R}\left(t-t^{\prime }\right)dt^{\prime }.$$
The sum of all $\varphi_{R}^{\left(n\right)}\left(t\right)$ gives the rate function of resetting events
$$\kappa \left(t\right)=\sum_{n=1}^{\infty }\varphi_{R}^{\left(n\right)}\left(t\right).$$
Taking the Laplace transform, defined as
$$\hat{f}\left(s\right)=L_{s}\left[f\left(t\right)\right]=\int_{0}^{\infty }f\left(t\right)e^{-st}dt$$
to the above equations, one obtains an expression for the resetting rate function in terms of the resetting PDF in Laplace space
(4)$$\hat{\kappa }\left(s\right)=\sum_{n=1}^{\infty }{\hat{\varphi }}_{R}^{\left(n\right)}\left(s\right)=\frac{{\hat{\varphi }}_{R}\left(s\right)}{1-{\hat{\varphi }}_{R}\left(s\right)}.$$
If the resetting is a Poissonian process (i.e., Markovian), then the times between consecutive resetting events are exponentially distributed [1,2,50]
(5)$$\varphi_{R}\left(t\right)=re^{-rt}$$
and from Equation (Equation 4), the rate at which resetting events follow is constant, κ(t)=r. However, if the times between resets are drawn from a power-law PDF [50,51,52], as
(6)$$\varphi_{R}\left(t\right)=\frac{\gamma r}{{\left(1+rt\right)}^{1+\gamma }},\;\gamma >0,$$
then the resetting process is non-Markovian and the rate of the resetting events depends on time. To compute this rate function, we first transform (Equation 6) to the Laplace space:(7)$${\hat{\varphi }}_{R}\left(s\right)=\gamma r\int_{0}^{\infty }\frac{e^{-st}}{{\left(1+rt\right)}^{1+\gamma }}dt=\gamma U\left(1,1-\gamma ,s/r\right),$$
and from Equation (Equation 4), the rate between resetting events in the Laplace is
(8)$$\begin{array}{c} \hat{\kappa }\left(s\right)=\frac{\gamma U(1,1-\gamma ,s/r)}{1-\gamma U(1,1-\gamma ,s/r)}, \end{array}$$
where U(a,b;z) is the Tricomi confluent hypergeometric function of the second kind [53]. This special function admits the following generalized power series expansion [53]
$$\begin{array}{cc} U\left(a,b,z\right)=\frac{\Gamma \left(1-b\right)}{\Gamma \left(1-b+a\right)} & \left(1+\frac{az}{b}+\ldots \right)\\ & +\frac{\Gamma \left(b-1\right)}{\Gamma \left(a\right)}z^{1-b}\left[1+\frac{(1+a-b)z}{2-b}+\ldots \right] \end{array}$$
for small *z*. Depending on the sign of *b* and the regions of its values, the leading terms are
(9)$$\begin{array}{c} U\left(a,b,z\right)=\left\{\begin{array}{cc} \frac{\Gamma \left(1-b\right)}{\Gamma \left(1-b+a\right)}\left(1+\frac{az}{b}+\ldots \right),\; & b<0,\\ \frac{\Gamma \left(1-b\right)}{\Gamma \left(1-b+a\right)}-\frac{\Gamma \left(b\right)}{\left(1-b\right)\Gamma \left(a\right)}z^{1-b}+\ldots ,\; & 0<b<1,\\ \frac{\Gamma \left(b-1\right)}{\Gamma \left(a\right)}\frac{1}{z^{b-1}}+\ldots ,\; & b>1. \end{array}\right. \end{array}$$
In particular, for s→0, which according to the Tauberian theorem corresponds to the long-time limit t→∞, one finds
(10)$$\begin{array}{c} 1-\gamma U\left(1,1-\gamma ,\frac{s}{r}\right)=\left\{\begin{array}{cc} \Gamma \left(1-\gamma \right){\left(\frac{s}{r}\right)}^{\gamma },\; & 0<\gamma <1,\\ \frac{s/r}{\gamma -1}.\; & \gamma >1. \end{array}\right. \end{array}$$
Plugging this result into Equation (Equation 8) and inverting by Laplace, the rate function has the long time asymptotic form
(11)$$\kappa \left(t\right)∼\left\{\begin{array}{cc} \frac{r}{\Gamma \left(\gamma \right)\Gamma \left(1-\gamma \right){\left(rt\right)}^{1-\gamma }},\; & 0<\gamma <1,\;\mathrm{slow}\;\mathrm{resetting},\\ (\gamma -1)r,\; & \gamma >1,\;\mathrm{fast}\;\mathrm{resetting}. \end{array}\right.$$
It can be shown that for 0<γ<1, the power-law PDF has diverging moments while for γ>1, the first moment exists, and for γ>2, the second moment exists as well. As can be seen from Equation (Equation 11) for 0<γ<1, the rate is decaying with time while for γ>1 it is constant. The resetting process is then faster for γ>1 than for 0<γ<1. We refer to the former as the *fast resetting* regime and the latter as the *slow resetting* regime. The existence of the first moment, i.e., a mean resetting time, is then equivalent to the existence of a constant resetting rate. Next, we illustrate how to find the MSD from the Langevin equations.

### 3.2. Resetting in Fingers

Let $P_{Y}^{0}\left(y,t\right)$ be the probability density that the walker is at point *y* of the finger at time *t* in the absence of resetting. When the movement of the walker along the fingers is affected by the resetting mechanism, then the resulting propagator for the motion in fingers follows the renewal master equation [54]
(12)$$P_{Y}\left(y,t\right)=\varphi_{R}^{*}\left(t\right)P_{Y}^{0}\left(y,t\right)+\int_{0}^{t}\varphi_{R}\left(t^{\prime }\right)P_{Y}\left(y,t-t^{\prime }\right)dt^{\prime },$$
where $\varphi_{R}^{*}\left(t\right)=\int_{t}^{\infty }\varphi_{R}\left(t^{\prime }\right)dt^{\prime }$ is the probability that the reset has not happened yet at time *t* (i.e., the resetting survival probability). The first term in the right-hand side accounts for the probability that no reset has happened until time *t*, in which case the propagator $P_{Y}^{0}\left(y,t\right)$ describes the motion. The second term accounts for the cases where at least one resetting event has occurred, after which the motion renews and it can be described by the overall propagator with a shift in time (see [54] for further details). Applying the Laplace transform to both sides of the equation and isolating the overall propagator, one obtains
(13)$${\hat{P}}_{Y}\left(y,s\right)=\frac{L_{s}\left[\varphi_{R}^{*}\left(t\right)P_{Y}^{0}\left(y,t\right)\right]}{1-{\hat{\varphi }}_{R}\left(s\right)}.$$
Introducing this result into the Laplace transform of Equation (Equation 3), the overall MSD in the Laplace space is
(14)$${\langle {\hat{X}}^{2}\left(s\right)\rangle }_{r}=2D_{x}\frac{L_{s}\left[\varphi_{R}^{*}\left(t\right)P_{Y}^{0}\left(y=0,t\right)\right]}{s[1-{\hat{\varphi }}_{R}\left(s\right)]},$$
which is general for any type of motion and reset time PDF in the fingers. To proceed further, we need to calculate the propagator $P_{Y}^{0}\left(y,t\right)$. To do this, we assume that the motion of the walker in fingers in the absence of resetting is described by a continuous time random walk whose propagator is given by the generalized diffusion equation [55]
(15)$$\frac{\partial P_{Y}^{0}}{\partial t}=\frac{\sigma^{2}}{2}\int_{0}^{t}K\left(t-t^{\prime }\right)\frac{\partial^{2}P_{Y}^{0}\left(y,t^{\prime }\right)}{\partial y^{2}}dt^{\prime },$$
where σ is the mean jump distance and the memory kernel K(t) is related to the waiting time PDF between jumps φ(t) through the relationship
(16)$$\hat{K}\left(s\right)=\frac{s\hat{\varphi }\left(s\right)}{1-\hat{\varphi }\left(s\right)}$$
in the Laplace space. If the walker is initially at y=0, then, from the Fourier–Laplace transform of (Equation 15), we find
(17)$${\tilde{P}}_{Y}^{0}\left(k,s\right)=\frac{1}{s+\frac{\sigma^{2}}{2}k^{2}\hat{K}\left(s\right)},$$
where $\tilde{F}\left(k,s\right)=\int_{0}^{\infty }dt\int_{-\infty }^{\infty }e^{-iky}e^{-st}F\left(y,t\right)dy$ stands for the Fourier–Laplace transform of the function F(y,t). Finally, ${\hat{P}}_{Y}^{0}\left(y=0,s\right)$ follows from the inverse Fourier transform
(18)$${\hat{P}}_{Y}^{0}\left(y=0,s\right)=\frac{1}{2\pi }\int_{-\infty }^{\infty }{\hat{P}}_{Y}^{0}\left(k,s\right)dk=\frac{1}{s\sigma \sqrt{2}}\sqrt{\frac{1}{\hat{\varphi }\left(s\right)}-1}.$$
Since we are interested in a walker moving subdiffusively along fingers, we consider that the waiting time takes the form [56]
(19)$$\hat{\varphi }\left(s\right)=\frac{1}{1+{\left(s\tau \right)}^{\alpha }},\;0<\alpha \leq 1,$$
in the Laplace space. The movement along the fingers described by Equation (Equation 15) is diffusive for α=1 and subdiffusive for 0<α<1. Substituting Equation (Equation 19) into Equation (Equation 18) and inverting by Laplace, we find
(20)$$P_{Y}^{0}\left(y=0,t\right)=\frac{1}{2\sqrt{D_{y}}\Gamma \left(1-\alpha /2\right)t^{\alpha /2}},$$
where we have defined the generalized diffusion coefficient $D_{y}=\sigma^{2}/\left[2\tau^{\alpha }\right]$. In the absence of resetting, the MSD can be found by inserting Equation (Equation 20) into Equation (Equation 3). This yields [29]
(21)$$\begin{array}{c} {\langle X^{2}\left(t\right)\rangle }_{0}=\frac{D_{x}}{\sqrt{D_{y}}}\frac{t^{1-\frac{\alpha }{2}}}{\Gamma \left(2-\frac{\alpha }{2}\right)}. \end{array}$$
In the presence of a resetting process, with reset times PDF $\varphi_{R}\left(t\right)$, the overall MSD follows from Equation (Equation 14)
(22)$${\langle {\hat{X}}^{2}\left(s\right)\rangle }_{r}=\frac{D_{x}}{\sqrt{D_{y}}\Gamma \left(1-\alpha /2\right)}\frac{L_{s}\left[\varphi_{R}^{*}\left(t\right)t^{-\alpha /2}\right]}{s[1-{\hat{\varphi }}_{R}\left(s\right)]}.$$To find the MSD of the walker’s motion through the comb given by Equation (Equation 22), we need to specify $\varphi_{R}\left(t\right)$. We consider below the cases of exponential and Pareto PDFs for $\varphi_{R}\left(t\right)$.

#### 3.2.1. Markovian Resetting

Considering the exponential PDF for resetting periods in (Equation 14), we find
(23)$${\langle {\hat{X}}^{2}\left(s\right)\rangle }_{r}=2D_{x}\frac{s+r}{s^{2}}{\hat{P}}_{Y}^{0}\left(y=0,s+r\right).$$In the long-time limit (s→0), this expression, after applying the inverse Laplace transform, reads
(24)$${\langle X^{2}\left(t\right)\rangle }_{r}∼2D_{x}\;{\hat{P}}_{Y}^{0}\left(y=0,r\right)rt,$$
which predicts a diffusive behavior as $\langle X^{2}\left(t\right)\rangle ∼t$. In consequence, the overall movement through the comb is diffusive if the resetting in fingers occurs at a constant rate. To check this general result with a specific example, let us assume that $P_{Y}^{0}\left(y,t\right)$ follows the generalized master Equation (Equation 15) where the waiting time PDF is given by Equation (Equation 19). Then, the MSD is given by Equation (Equation 22) and can be found to be
(25)$${\langle {\hat{X}}^{2}\left(s\right)\rangle }_{r}=\frac{D_{x}}{\sqrt{D_{y}}}\frac{{\left(s+r\right)}^{\alpha /2}}{s^{2}}.$$Taking the inverse Laplace transform, we find the overall MSD through the comb of a walker moving diffusively or subdiffusively along the fingers under exponential resetting; this is
(26)$$\begin{array}{c} {\langle X^{2}\left(t\right)\rangle }_{r}=\frac{D_{x}}{\sqrt{D_{y}}}L_{t}^{-1}\left[\frac{s^{-2}}{{\left(s+r\right)}^{-\alpha /2}}\right]=\frac{D_{x}}{\sqrt{D_{y}}}t^{1-\alpha /2}E_{1,2-\alpha /2}^{-\alpha /2}\left(-rt\right), \end{array}$$
where $E_{\mu ,\beta }^{\gamma }\left(z\right)=\sum_{n=0}^{\infty }\frac{{\left(\gamma \right)}_{n}}{\Gamma (\mu n+\beta )}\frac{z^{n}}{n!}$ is the three-parameter Mittag–Leffler function [57] with the Pochhammer symbol ${\left(\gamma \right)}_{n}=\Gamma \left(\gamma +n\right)/\Gamma \left(n\right)$. The MSD has the following asymptotic form (The asymptotic behavior of the three-parameter Mittag–Leffler function is given by $E_{\mu ,\beta }^{\gamma }\left(-{\lambda t}^{\mu }\right)∼\frac{{\left({\lambda t}^{\mu }\right)}^{-\gamma }}{\Gamma \left(\beta -\mu \gamma \right)}$ for $\lambda t^{\mu }\gg 1$ and 0<μ<2, see Ref. [58])
(27)$${\langle X^{2}\left(t\right)\rangle }_{r}∼\frac{D_{x}r^{\alpha /2}}{\sqrt{D_{y}}}t\;\mathrm{as}\;t\to \infty .$$We note that even if the walker moves subdiffusively (with 0<α<1) along the fingers, the resetting process is so fast that the walker is driven to the backbone at a constant rate. This breaks down the trapping effect of the motion along the fingers in the absence of resetting and the overall MSD is diffusive. As shown in Equation (Equation 24), when the resetting process to the backbone is Markovian, the overall MSD is diffusive regardless of the specific movement along the fingers.

#### 3.2.2. Non-Markovian Resetting

When the resetting times PDF follows the power-law function in Equation (Equation 6), the resetting process is non-Markovian. In addition, if the motion along the fingers is subdiffusive, then the overall MSD can be computed from (Equation 22) and is expressed as
(28)$${\langle {\hat{X}}^{2}\left(s\right)\rangle }_{r}=\frac{D_{x}}{r^{1-\alpha /2}\sqrt{D_{y}}}\frac{U(1-\alpha /2,2-\gamma -\alpha /2,s/r)}{s[1-\gamma U(1,1-\gamma ,s/r)]}.$$Now, considering the results given by Equation (Equation 9), the Tricomi function in the numerator of Equation (Equation 28) reads
(29)$$\begin{array}{cc} U & \left(1-\frac{\alpha }{2},2-\gamma -\frac{\alpha }{2},\frac{s}{r}\right)\\ & =\left\{\begin{array}{cc} \frac{\Gamma \left(1-\gamma -\frac{\alpha }{2}\right)}{\Gamma \left(1-\frac{\alpha }{2}\right)}\frac{1}{{\left(s/r\right)}^{1-\gamma -\frac{\alpha }{2}}}+\ldots ,\; & 0<\gamma <1-\frac{\alpha }{2},\\ \frac{\Gamma \left(-1+\gamma +\frac{\alpha }{2}\right)}{\Gamma \left(\gamma \right)}-\frac{\Gamma \left(2-\gamma -\frac{\alpha }{2}\right)}{\Gamma \left(1-\frac{\alpha }{2}\right)\left(-1+\gamma +\frac{\alpha }{2}\right)}{\left(s/r\right)}^{-1+\gamma +\frac{\alpha }{2}}+\ldots ,\; & 1-\frac{\alpha }{2}<\gamma <2-\frac{\alpha }{2},\\ \frac{\Gamma \left(-1+\gamma +\frac{\alpha }{2}\right)}{\Gamma \left(\gamma \right)}\left(1+\frac{1-\frac{\alpha }{2}}{2-\gamma -\frac{\alpha }{2}}\frac{s}{r}+\ldots \right),\; & \gamma >2-\frac{\alpha }{2}, \end{array}\right. \end{array}$$
for s→0. Inserting the above expressions and Equation (Equation 10) in Equation (Equation 28) and inverting by Laplace, we get the behavior of the overall MSD in the long-time limit (t→∞)
(30)$$\begin{array}{ccc} {\langle X^{2}\left(t\right)\rangle }_{r} & ∼\frac{D_{x}}{\sqrt{D_{y}}\;r^{1-\alpha /2}}\frac{\Gamma \left(1-\frac{\alpha }{2}-\gamma \right)}{\Gamma \left(1-\gamma \right)\Gamma \left(1-\frac{\alpha }{2}\right)}\frac{{\left(rt\right)}^{1-\alpha /2}}{\Gamma \left(2-\frac{\alpha }{2}\right)},\; & 0<\gamma <1-\frac{\alpha }{2},\\ {\langle X^{2}\left(t\right)\rangle }_{r} & ∼\frac{D_{x}}{\sqrt{D_{y}}\;r^{1-\alpha /2}}\frac{\Gamma \left(-1+\frac{\alpha }{2}+\gamma \right)}{\Gamma \left(1-\gamma \right)\Gamma \left(\gamma \right)}\frac{{\left(rt\right)}^{\gamma }}{\Gamma \left(1+\gamma \right)},\; & 1-\frac{\alpha }{2}<\gamma <1.\\ {\langle X^{2}\left(t\right)\rangle }_{r} & ∼\frac{D_{x}}{\sqrt{D_{y}}\;r^{1-\alpha /2}}\frac{\Gamma \left(-1+\frac{\alpha }{2}+\gamma \right)}{\Gamma \left(\gamma -1\right)}rt,\; & \gamma >1. \end{array}$$Notably, for power-law resetting time PDFs, the asymptotic limit of the overall MSD depends explicitly on the characteristic exponents γ and α of the resetting PDF and the waiting time PDF in fingers, respectively. Now, we relate the temporal scaling in (Equation 30) with the rate of the resetting events. The effect of the heavy-tailed waiting time PDF is to keep the walker moving along the fingers. On the other hand, the resetting process pushes the walker towards the backbone. This interesting interplay is shown in the exponent of the expressions of the MSD in Equation (Equation 30). When γ is small, the resetting process is slow and the dynamics of the walker are dominated by the waiting time PDF and the walker behaves as in the absence of resetting. Note that in this case, the scaling dependence of the MSD in Equation (Equation 21) is the same as in the first equation of (Equation 30). When γ>1, the resetting process is fast and it dominates the dynamics, and the walker visits the backbone very frequently. Since the motion along the backbone is assumed to be diffusive, in this case the walker also moves diffusively. This is in agreement with temporal scaling of the third equation of (Equation 30). Finally, an interesting intermediate case appears when 1−α/2<γ<1. In this case, the temporal scaling of the MSD is subdiffusive but, since the exponent γ is higher than the exponent 1−α/2 corresponding to the slow resetting case, the overall movement is subdiffusive.

In Figure 2, we plot the results for the MSD obtained from the numerical inverse Laplace transform of (Equation 25) and (Equation 28), which is implemented in Mathematica [59]. The results in the long-time limit correspond to the asymptotic behavior of the MSDs given by (Equation 27) and (Equation 30), respectively. From the graphics, it is evident that in the short-time limit the MSD behaves as the MSD in the case of no resetting, i.e., there is no dependence on the resetting parameters *r* and γ, but only on α, as given by (Equation 21).

### 3.3. Global Resetting

Here, we analyze what happens when resetting is global, meaning that in every moment the particle may return to the origin and start anew, regardless of whether it is located at the fingers or at the backbone. In the case of resetting in fingers, the walker is forced to move to the point of the backbone located at the finger’s position and then the walker may choose to move along the backbone driven by the noise $\xi_{x}$ performing a Brownian motion or enter in the fingers and move there. However, when the resetting is global, the walker is forced to move to the origin wherever it is. Then, it is expected that in this case the resetting process has a stronger effect on the MSD scaling than if resetting occurs in the fingers only. Now, we can derive an equation for the overall MSD that can be found in terms of the MSD without resets (see [54] for further details). It reads
(31)$${\langle {\hat{X}}^{2}\left(s\right)\rangle }_{r}=\frac{L_{s}\left[\varphi_{R}^{*}\left(t\right){\langle X^{2}\left(t\right)\rangle }_{0}\right]}{1-{\hat{\varphi }}_{R}\left(s\right)},$$
where ${\langle X^{2}\left(t\right)\rangle }_{0}$ is the MSD of the motion in the comb without resetting. Inserting Equation (Equation 3) into (Equation 31), we obtain
(32)$${\langle {\hat{X}}^{2}\left(s\right)\rangle }_{r}=2D_{x}\frac{L_{s}\left[\varphi_{R}^{*}\left(t\right)\int_{0}^{t}P_{Y}^{0}\left(y=0,t^{\prime }\right)dt^{\prime }\right]}{1-{\hat{\varphi }}_{R}\left(s\right)}.$$If the motion in the fingers is the same as in the previous section, then using (Equation 20), Equation (Equation 32) turns into
(33)$${\langle {\hat{X}}^{2}\left(s\right)\rangle }_{r}=\frac{D_{x}}{\sqrt{D_{y}}\Gamma \left(2-\frac{\alpha }{2}\right)}\frac{L_{s}\left[\varphi_{R}^{*}\left(t\right)t^{1-\alpha /2}\right]}{1-{\hat{\varphi }}_{R}\left(s\right)}.$$As we did in the previous section, below we study the two cases (Markovian and non-Markovian) where the resetting times are drawn from an exponential or power-law PDF.

#### 3.3.1. Markovian Resetting

As in the previous section, we do not need to specify the motion along the fingers. Considering (Equation 5) in (Equation 32), the overall MSD can be written as
(34)$$\begin{array}{c} {\langle {\hat{X}}^{2}\left(s\right)\rangle }_{r}=2D_{x}\frac{{\hat{P}}_{Y}^{0}\left(y=0,s+r\right)}{s}. \end{array}$$Taking the long-time limit s→0 and inverting by Laplace, we find that the overall MSD
(35)$$\begin{array}{c} {\langle X^{2}\left(t\right)\rangle }_{r}∼2D_{x}\;{\hat{P}}_{Y}^{0}\left(y=0,s=r\right),\;\mathrm{as}\;t\to \infty . \end{array}$$Therefore, in the case of exponentially distributed resets, the overall MSD reaches a stationary value which depends explicitly on the probability of being in the backbone. In particular, if the propagator for the motion along the fingers follows the generalized diffusion Equation (Equation 15), then
(36)$$\begin{array}{c} {\langle X^{2}\left(t\right)\rangle }_{r}∼\frac{D_{x}}{\sqrt{D_{y}}\;r^{1-\alpha /2}}\;\mathrm{as}\;t\to \infty . \end{array}$$Notably, in the case of global resetting, its effect is stronger than in the case of resetting in fingers. This is due to the fact that in the latter case the walker is reset to a static point while in the former case the walker is reset to a backbone where it can keep on moving diffusively.

#### 3.3.2. Non-Markovian Resetting

In the case when the resetting is described by the power-law PDF in Equation (Equation 6), the expression for the overall MSD in the Laplace space can be directly derived from Equation (Equation 33):(37)$${\langle {\hat{X}}^{2}\left(s\right)\rangle }_{r}=\frac{D_{x}}{\sqrt{D_{y}}\;r^{2-\alpha /2}}\frac{U(2-\alpha /2,3-\gamma -\alpha /2,s/r)}{1-\gamma U(1,1-\gamma ,s/r)}$$
in terms of the Tricomi confluent hypergeometric functions. Again, for these functions we analyze the small *s* limit using the relations (Equation 9). In particular,
(38)$$\begin{array}{cc} U & \left(2-\frac{\alpha }{2},3-\gamma -\frac{\alpha }{2},\frac{s}{r}\right)\\ & =\left\{\begin{array}{cc} \frac{\Gamma \left(2-\gamma -\frac{\alpha }{2}\right)}{\Gamma \left(2-\frac{\alpha }{2}\right)}\frac{1}{{\left(s/r\right)}^{2-\gamma -\frac{\alpha }{2}}}+\ldots ,\; & 0<\gamma <2-\frac{\alpha }{2},\\ \frac{\Gamma \left(-2+\gamma +\frac{\alpha }{2}\right)}{\Gamma \left(\gamma \right)}-\frac{\Gamma \left(3-\gamma -\frac{\alpha }{2}\right)}{\Gamma \left(2-\frac{\alpha }{2}\right)\left(-2+\gamma +\frac{\alpha }{2}\right)}{\left(s/r\right)}^{-2+\gamma +\frac{\alpha }{2}}+\ldots ,\; & 2-\frac{\alpha }{2}<\gamma <3-\frac{\alpha }{2},\\ \frac{\Gamma \left(-2+\gamma +\frac{\alpha }{2}\right)}{\Gamma \left(\gamma \right)}\left(1+\frac{2-\frac{\alpha }{2}}{3-\gamma -\frac{\alpha }{2}}\frac{s}{r}+\ldots \right),\; & \gamma >3-\frac{\alpha }{2}. \end{array}\right. \end{array}$$Introducing (Equation 10) and (Equation 38) in (Equation 37) and inverting by Laplace, one finally finds the overall MSD in the long-time limit
(39)$$\begin{array}{ccc} {\langle X^{2}\left(t\right)\rangle }_{r} & ∼\frac{D_{x}}{\sqrt{D_{y}}\;r^{1-\alpha /2}}\frac{\Gamma \left(2-\frac{\alpha }{2}-\gamma \right)}{\Gamma \left(1-\gamma \right)\Gamma {\left(2-\frac{\alpha }{2}\right)}^{2}}{\left(rt\right)}^{1-\alpha /2},\; & 0<\gamma <1,\\ {\langle X^{2}\left(t\right)\rangle }_{r} & ∼\frac{D_{x}}{\sqrt{D_{y}}\;r^{1-\alpha /2}}\frac{\gamma -1}{\Gamma \left(2-\frac{\alpha }{2}\right)\left(2-\gamma -\alpha /2\right)}{\left(rt\right)}^{2-\gamma -\alpha /2},\; & 1<\gamma <2-\frac{\alpha }{2},\\ {\langle X^{2}\left(t\right)\rangle }_{r} & ∼\frac{D_{x}}{\sqrt{D_{y}}\;r^{1-\alpha /2}}\frac{\Gamma \left(-2+\frac{\alpha }{2}+\gamma \right)}{\Gamma \left(\gamma -1\right)},\; & \gamma >2-\frac{\alpha }{2}. \end{array}$$Again, for power-law reset time PDFs, the asymptotic limit of the overall MSD depends explicitly on γ and α. The interplay between the tails of the power-law reset time and the waiting time PDFs is shown again in the exponent of the expressions for the MSD in Equation (Equation 39). When 0<γ<1, the resetting is slow and the dynamics of the walker are dominated by the waiting time PDF and, analogously as in the case of resetting in fingers, the walker behaves as in the absence of resetting; the MSD is subdiffusive. When γ>2−α/2, the resetting process is very fast and it dominates the dynamics of the walker. Since the reset displaces the walker to the origin, the stochastic localization emerges and the overall MSD reaches a constant value. Finally, when 1<γ<2−α/2, the exponent of the MSD is lower than 1−α/2, i.e, it is subdiffusive. We note that the obtained results are in agreement with those obtained in [52], where α used in the paper corresponds to 1−α/2 used in the present work. We also note that in the limiting cases γ=1 and γ=2−α/2, one observes logarithmic behavior of the MSD, see Ref. [52].

In Figure 3, we plot the results for the MSD obtained from numerical inverse Laplace transform of (Equation 34) (left panel) and (Equation 37) (right panel). The results in the long-time limit correspond to the asymptotic behavior of the MSDs given by (Equation 36) and (Equation 39), respectively. In the left panel, we see that the MSD in all cases tends to a constant value in agreement with the scaling behavior predicted in Equation (Equation 35). In the right panel, we distinguish the three cases predicted in Equation (Equation 39). Here, one observes that in the short-time limit the MSD behaves as the MSD in the case of no resetting, i.e., there is only dependence on α.

## 4. Conclusions

We have computed the MSD of a walker moving on a comb-like structure under the effect of a stochastic (Markovian and non-Markovian) resetting. By using a set of Langevin equations, we obtain the expression for the MSD in terms of the reset times PDF and the propagator of the motion along the fingers when either the reset takes place at the fingers (Equation (Equation 14)) or when it is global (Equation (Equation 32)). We have assumed that the movement along fingers is described by a continuous time random walk with a power-law waiting time PDF with exponent α, while the movement along the backbone is diffusive. When the resetting process is Markovian, i.e., the reset times are drawn from a exponential PDF, then the long-time limit of the overall MSD is either diffusive or constant when the reset takes place in fingers or it is global, respectively. When the reset times PDF is a power-law PDF with exponent γ, then an interesting interplay emerges between its tail and that of the waiting time PDF. For resetting in fingers, the MSD may be diffusive or subdiffusive in the long-time limit. When γ>1, it is diffusive and subdiffusive otherwise. The exponent of the MSD in the subdiffusive case depends on α but the critical value of γ for the transition between diffusion and subdiffusion does not. For the case of global resetting, the MSD may be constant or subdiffusive in the long-time limit. When γ>2−α/2, it is constant (stochastic localization) and subdiffusive otherwise. In this case, the critical value of γ for the transition between stochastic localization and subdiffusion depends explicitly on α. These results on non-Markovian (power-law) resetting are summarised in Table 1. Calculation of the time averaged MSD for the considered comb model could be of interest for future investigation. This can be analyzed by using the approach presented in Refs. [10,13]. Future research could also be related to the investigation of random walks on comb structures in the presence of time-dependent [60] and non-instantaneous resettings [61], random walks on combs in the presence of resetting in an interval [62,63,64] and bounded in complex potential [65], discrete space–time resetting models [15] for comb structures, and finite-velocity diffusion processes on comb [66,67,68,69,70] with non-Markovian resetting.

## Figures and Tables

**Figure 1 entropy-25-01529-f001:**
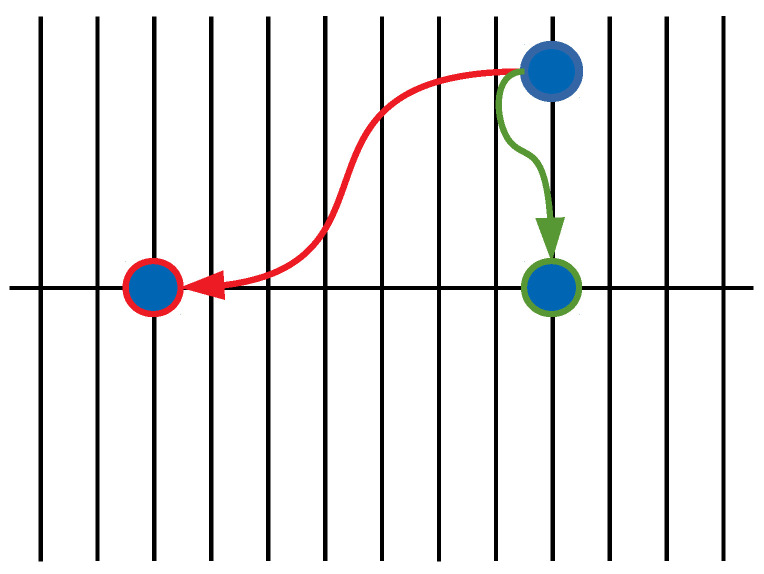
Graphical representation of resetting mechanisms on comb: (i) return to the backbone (resetting in fingers)—green line; (ii) return to the origin (global resetting)—red line.

**Figure 2 entropy-25-01529-f002:**
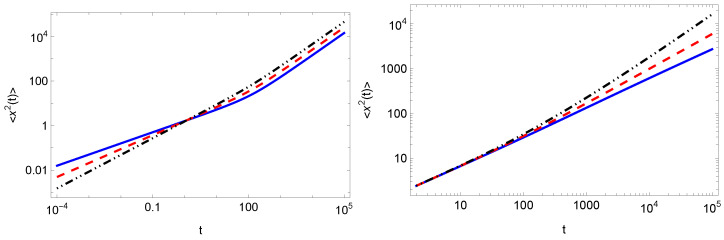
MSD in the case of Markovian resetting computed from the numerical inversion of the Laplace transform of Equation (Equation 25) (left panel) for r=0.01 and α=1 (blue solid line), α=3/4 (red dashed line), α=1/2 (black dot-dot-dashed line) and non-Markovian resetting (right panel) computed from the numerical inversion of the Laplace transform of Equation (Equation 28) for r=0.01, α=3/4 and γ=1/4 (blue solid line), γ=3/4 (red dashed line), γ=3/2 (black dot-dot-dashed line). We also use $D_{x}=1$ and $D_{y}=1/2$. In the left panel, we see that in the long-time limit all lines are parallel in agreement with the linear predicted by scaling Equation (Equation 27). However, in the right panel according to Equation (Equation 30), the slopes are different because in this case the exponents of *t* depend on α.

**Figure 3 entropy-25-01529-f003:**
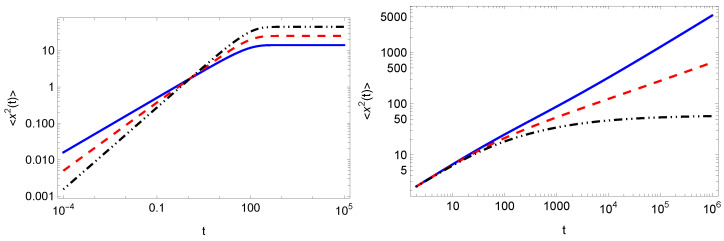
MSD in the case of Markovian resetting (left panel) for r=0.01 and α=1 (blue solid line), α=3/4 (red dashed line), α=1/2 (black dot-dot-dashed line). We see that the MSD in all cases tends to a constant value in agreement with the scaling behavior predicted in Equation (Equation 35). For non-Markovian resetting (right panel), we take r=0.01, α=3/4 and γ=1/2 (blue solid line), γ=5/4 (red dashed line), γ=2 (black dot-dot-dashed line). We also use $D_{x}=1$ and $D_{y}=1/2$. The results have been obtained by numerically inverting Equations (Equation 34) (left panel) and (Equation 37) (right panel).

**Table 1 entropy-25-01529-t001:** Long-time behavior of the MSD along the backbone in the presence of non-Markovian resetting.

	Resetting in Fingers	Global Resetting
$0<\gamma <1-\frac{\alpha }{2}$	${\langle X^{2}\left(t\right)\rangle }_{r}∼t^{1-\alpha /2}$	${\langle X^{2}\left(t\right)\rangle }_{r}∼t^{1-\alpha /2}$
$1-\frac{\alpha }{2}<\gamma <1$	${\langle X^{2}\left(t\right)\rangle }_{r}∼t^{\gamma }$	${\langle X^{2}\left(t\right)\rangle }_{r}∼t^{1-\alpha /2}$
$1<\gamma <2-\frac{\alpha }{2}$	${\langle X^{2}\left(t\right)\rangle }_{r}∼t$	${\langle X^{2}\left(t\right)\rangle }_{r}∼t^{2-\gamma -\alpha /2}$
$2-\frac{\alpha }{2}<\gamma$	${\langle X^{2}\left(t\right)\rangle }_{r}∼t$	${\langle X^{2}\left(t\right)\rangle }_{r}∼\mathrm{const}$

## Data Availability

No new data were created or analyzed in this study. Data sharing is not applicable to this article.

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
