# Peer review of "Random Walks on Comb-like Structures under Stochastic Resetting"

_entropy, 2023, doi:10.3390/e25111529_

Round 1
Reviewer 1 Report
Comments and Suggestions for Authors
I have reviewed the manuscript titled "Random walks on comb-like structures under stochastic
resetting" by Masó-Puigdellosas and et al. The authors investigated the long-time dynamics of the mean squared displacement of a random walker
moving on a comb structure under two types of stochastic resetting.
The manuscript is well written and structured.
Overall, this is an interesting work deserving publication though a few minor suggestions are listed below.
1. The topic of stochastic reset is not adequately introduced to the readers in Section 1.
2. The MSD at short times appears to be straightward because the reset almost has no effect when then time is well shorter than the reset time scale. If so, in fig 1 and 2, the MSD at short time should have scaling $t^{1-\alpha/2}$.
Reviewer 2 Report
Comments and Suggestions for Authors​The current manuscript considers a problem of stochastic resetting for the process of particle diffusion on a comb structure. The material is very well-written and---being mathematically intricate---perfectly fits the scope of the journal. Prior to its acceptance, the authors are encouraged to address the points listed below.
The comb model is definitely nice, but it is rather old. The authors are thus encouraged to find and to list concrete applications of the setup they propose, with two types of resetting mechanisms and in the presence of anomalous diffusion along the “fingers” in the structure. Such real-world examples are important to justify a very specific model the authors propose.
Despite the age of comb-type models, the authors could still include a picture of a "tree" with multiple “branches” representing the fingers, and a particle diffusing with reset events on such a structure. Here, a couple of questions can be discussed. What is the distribution of branches from the bottom of the tree (the position of the global-resetting point)? Would the model predict different results for different distributions of branches along the stem (say, branches growing infinitely high with a uniform density, or branches forming a finite-size tree crown, being distributed from the bottom to the top say with a Rayleigh-type distribution)? Along the same lines, can one incorporate a (general) distribution of branch lengths into the current model?
The authors should concretely indicate what are the novel results of the current study as compared to their previous works on a very similar subject (see refs. 5, 6, and 20), apart from subdiffusion in the comb "fingers".
Applications of the proposed model with subdiffusion and resetting should be explicitly mentioned. Without such applications, the current material looks as a straightforward generalization of some previous studies of the authors. Why do we need these two distinct types of resettings? Also, how do the authors imagine a resetting from deeply visited dead-end "fingers" to the initial point of the motion? Can this process be connected to a finite life-time of diffusing particles, for instance?
In the resetting literature, the Poissonian distribution for the waiting times of resetting is very popular. How universal are the results obtained within such a constant-resetting-rate propagation scheme? How realistic is the exponential resetting for real physical systems involving comb structures with resetting?
If the authors cite ref. [10], then ref. [​DOI: 10.1103/PhysRevE.104.024105​]---which is more general study also containing the TAMSD-results for the sam​e process---should also be mentioned. The same reasoning applies to ref. [9] and its subsequent generalization published as ref. [DOI: 10.1103/Ph​ysRevE.106.034137]. Note also that a model similar to reset GBM has been known long ago in the theory of catastrophes, see and mention e.g. refs. [doi:10.2307/1427051], [https://doi.org/10.1016/0167-7152(94)90048-5], and [https://doi.org/10.1080/15326349708807425].
In eq. 9 not only the sign of parameter b, but also the regions of its values are important: please adjust the text appropriately.
Prior to eq. 24: regarding ", after inversion by Laplace, ": i doubt this person is still alive, so please reformulate the sentence.
The discussion around fig. 1 focuses on multiple regions and describes several regimes of the short- and long-time behaviors of the MSDs and their exponents. For reader’s convenience, a 2D figure illustrating all different regimes and their scaling exponents depending on the values of parameters alpha and beta would be very useful here. As an example of such a plot, please see e.g. fig. 2 in ref. [DOI: https://doi.org/10.1103/PhysRevE.101.062117]. T​his way, the discussion around eqs. 30 and 39 can be simplified and possibly also shortened.
The conclusions section should be extended; one can use the table with exponents from the previous comment to make it more systematic. Additionally, the authors only studied the properties of the MSD. It is clearly an important characteristics of the dynamics; but not the only important quantifier. For instance, the TAMSD can deliver additional important features (see the 2021 and 2022 PREs mentioned above), particularly valuable for the analysis of single-particle trajectories from the experiments. The MSD, instead, examines the spreading properties in a large ensemble of particles.
Reviewer 3 Report
Comments and Suggestions for Authors
This manuscript deals with the problem of a particle diffusing on a comb-structure under resetting. Diffusion along the backbone is assumed to be normal, whereas it is assumed to be subdiffusive along the fingers of the backbone. Two different types of resetting are considered, namely, local resetting (from anywhere in a finger to its intersection point with the backbone) and global resetting. In both cases, the subcases of Poissonian and non-Markovian resetting are considered.
A Markovian resetting from the fingers is found to induce normal diffusive motion, thereby minimizing the trapping effect of fingers. In contrast, for non-Markovian local resetting, an interesting crossover with three different regimes emerges, two of them diffusive and one of them normal diffusive. Thus, an interesting interplay between the exponents characterizing the waiting time distributions of the subdiffusive random walk and resetting takes place. As for global resetting, its effect is even more drastic, as it precludes normal diffusion. Specifically, such a resetting can induce a constant asymptotic MSD in the Markovian case or two distinct regimes of subdiffusive motion in the non-Markovian case.
In spite of the wealth of existing literature on comb models, the results are undoubtedly of sufficient interest to merit publication in Entropy because of the interesting interplay described above. The presentation is also clear and well written. Having said this, I do have a few comments and suggestions to improve the manuscript that the authors may wish to consider.
1. The study of the model is justified a posteriori by the interesting outcome, but I wonder if there is any real system that might be close to it. If so, it would definitely be an added value to mention it in the Introduction or where it is deemed to be pertinent.
2. The competition between diffusion in the y-direction and resetting has as a result making the propagation along the backbone more or less effective, since the trapping time in the fingers is controlled by the variables of these two processes. On the other hand, it is implicitly assumed that the length of each finger is infinite. However, as the authors surely know, models with fingers of different lengths have been considered, e.g. one in which the finger lengths are drawn from a power-law distribution [see e.g. S. Havlin and D. ben-Avraham, Adv. Phys. 36, 695 (1987) and S. Havlin, J. E. Kiefer, and G. H. Weiss, Phys. Rev. A 36, 1403 (1987), apart from more recent references]. Here, as a result of the impact on the trapping time distribution in fingers, a (simpler) crossover between subdiffusive and normal backbone motion occurs. The authors may find it pertinent to comment on this.
3. The abstract gives no hint about the obtained results, I would recommend to anticipate a summary of them (at the expense of losing some suspense!). It is also not clear from the text in the abstract whether both types of resetting act simultaneously or (as it is the case) they are considered separately. Please disambiguate.
4. By analogy with the notation for Dy, the notation Dx reminds one of a diffusion coefficient, but it is not. I would recommend to change this. On the other hand, the authors may wish to remind the reader that Dy does depend on alpha.
5. PY0(y,t) is defined as a probability in 3.2, but it is actually a pdf of dimension length-1. I recommend to correct this.
5. There is a misprint in the fourth-to-last line of the Conclusions: The case of global diffusion-> The case of global resetting.
Comments on the Quality of English LanguageEnglish is fine. However, a few sentences need minor editing, e.g.,
"Here we consider global and resetting in fingers of walker moving diffusively along the backbone but subdiffusively when it moves along the fingers."
->Here we consider global and local resetting (in fingers) of a walker moving diffusively along the backbone but subdiffusively along the fingers."
"regardless if it is located..."->"regardless of whether it is located..."
Please give the manuscript one last reading after any minor amendment is finished.
Round 2
Reviewer 2 Report
Comments and Suggestions for Authors
A satisfactory revision by the authors: the material deserves publication in the present form.